# Genome-Wide DNA Methylation Profile Indicates Potential Epigenetic Regulation of Aging in the Rhesus Macaque Thymus

**DOI:** 10.3390/ijms232314984

**Published:** 2022-11-29

**Authors:** Hong Qiu, Haobo Li, Ruiwen Fan, Yang Song, Xuan Pan, Chunhui Zhang, Jing Li

**Affiliations:** Key Laboratory of Bio-Resources and Eco-Environment of Ministry of Education, College of Life Sciences, Sichuan University, Chengdu 610064, China

**Keywords:** DNA methylation, mRNA expression, thymus, aging, ASM

## Abstract

We analyzed whole-genome bisulfite sequencing (WGBS) and RNA sequencing data of two young (1 year old) and two adult (9 years old) rhesus macaques (*Macaca mulatta*) to characterize the genomic DNA methylation profile of the thymus and explore the molecular mechanism of age-related changes in the thymus. Combining the two-omics data, we identified correlations between DNA methylation and gene expression and found that DNA methylation played an essential role in the functional changes of the aging thymus, especially in immunity and coagulation. The hypomethylation levels of *C3* and *C5AR2* and the hypermethylation level of *C7* may lead to the high expressions of these genes in adult rhesus macaque thymuses, thus activating the classical complement pathway and the alternative pathway and enhancing their innate immune function. Adult thymuses had an enhanced coagulation pathway, which may have resulted from the hypomethylation and upregulated expressions of seven coagulation-promoting factor genes (*F13A1*, *CLEC4D*, *CLEC4E*, *FCN3*, *PDGFRA*, *FGF2* and *FGF7*) and the hypomethylation and low expression of *CPB2* to inhibit the degradation of blood clots. Furthermore, the functional decline in differentiation, activation and maturation of T cells in adult thymuses was also closely related to the changes in methylation levels and gene expression levels of T cell development genes (*CD3G*, *GAD2*, *ADAMDEC1* and *LCK*) and the thymogenic hormone gene *TMPO*. A comparison of the age-related methylated genes among four mammal species revealed that most of the epigenetic clocks were species-specific. Furthermore, based on the genomic landscape of allele-specific DNA methylation, we identified several age-related clustered sequence-dependent allele-specific DNA methylated (cS-ASM) genes. Overall, these DNA methylation patterns may also help to assist with understanding the mechanisms of the aging thymus with the epigenome.

## 1. Introduction

As one of the key immunological organs, the thymus is an essential organ for the generation of the adaptive immune system and involves the differentiation and maturation of T lymphocytes (T cells) [1]. The thymus is also an organ of the endocrine system and by using the thymic hormone and thymosin, it regulates immune function [2]. The organ and thymic hormone have synergistic or antagonistic actions within several endocrine glands [2]. In most mammals, a peculiar feature of the thymus is that it shrinks or disappears as an individual gets older. The thymus starts deteriorating after birth, and thymic cells are eventually replaced by adipocytes [3,4]. Along with the age-related decrease in size and activity of the thymus comes an associated functional decline [5,6]. However, the molecular regulation mechanism of thymus development and aging remains unresolved.

As an important epigenetic modification, DNA methylation plays vital roles in cell differentiation, tissue-specific gene expression, genomic imprinting and X chromosome inactivation in both animals and plants. DNA methylation mainly regulates gene silencing or expression through the depletion of methylated sites, such as the cytosine-phosphate-guanine dinucleotide (CpG) of the formation of 5-methylcytosine (5 mC) [7]. DNA modification by 5 mC is involved in various ways in the development and aging process of various tissues and organs. Previous studies found that changes in DNA methylation levels in different functional regions (such as the promoter regions and differentially methylated regions (DMRs)) and chromosomal methylation imbalances are closely associated with the development and aging process [8,9,10]. Hypermethylation at promoter CpG islands typically results in the decreased transcription of downstream genes [10,11]. Allelic differentially methylated regions (aDMRs) or clustered S-ASM (cS-ASM) are closely associated with the preferential silencing of one of the alleles on homologous chromosomes [12,13,14,15,16]. How DNA methylation is involved in gene expression in the thymus with aging is poorly studied.

Based on findings from physiology and histology studies, the thymus’ immune function changes with age. The activity of the complement system increases and coagulation function enhances with age while hematopoietic ability decreases [17,18,19]. Previous research also found that epigenetic alterations represent one crucial mechanism behind the deteriorated cellular functions observed during aging and in age-related disorders [20,21,22]. In this study, we built the genomic DNA methylation profile of the thymus and analyzed two-omics data of the thymus at different ages. By comparing the two-omics data (methylome data and transcriptome data), we found dynamic changes in DNA methylation and gene expression during aging, immunity function changes via DNA methylation regulation and specific ASMs at two ages.

## 2. Results

### 2.1. WGBS Data Processing and Alignment

We obtained WGBS data of the thymus from four rhesus macaques. The bisulfite conversion rates for control samples ranged from 99.57% to 99.88%; the median was 99.73%. We achieved sequencing depths ranging between 35.40 and 37.29× (Appendix A). The number of paired-end alignments with a unique best hit ranged from 270,792,903 to 301,019,076. The mapping efficiency ranged from 94.25% to 95.83% with a median of 95.18%. The ratio of methylated cytosines in the total cytosines was 3.11% to 4.06%. The ratio of methylated cytosines in the CpG context in the total methylated cytosines (5 mC) ranged from 84.68% to 92.00%. Despite the appearance of the unknown 5 mC point, the known 5 mC sites were matched to three contexts at least 94.27% of the time (Appendix A). Then, we achieved a total count of deduplicated leftover sequences ranging between 215,049,201 (78.26% of the total) and 247,903,543 (85.67% of the total) (Appendix A). The average methylation level per sample in the CpG context of 5 mC was at least 0.7 (Appendix A).

### 2.2. DNA Methylation Profiles of the Thymus in Rhesus Macaques

We performed a principal component analysis (PCA) and Spearman’s correlation analyses to estimate the influences of individual variations. DNA methylation levels of four different genomic regions of 29,324 autosomal genes were analyzed. PCA analysis (Figure 1A) showed that PC1 accounting for 55.9% of the total variation was based on the genome regions. However, within each genomic region, we noted that samples from the same age were obviously grouped closer, which indicated that age contributed to the methylation level of all four regions. The Spearman’s correlation analyses (Figure 1B) showed correlation coefficients for relationships between all variables, including the categories of genomic regions, age and gender. The primary clustering was based on the categories of genomic regions. Overall, the methylation level was determined by the genomic region, followed by age and gender.

Three context categories (CHH, CHG and CpG) were measured by at least five reads per site in all samples. The number of CHH contexts ranged from 478,494,921 to 694,464,684, which was much higher than for CHG and CpG. Meanwhile, the CpG context had the lowest number ranging from 26,482,621 to 42,576,680 (Appendix A). However, the CpG contexts showed the highest methylation levels (close to 1) among the three categories; in contrast, the methylation levels of CHH and CHG were both close to 0 (Appendix A). When statistically analyzing by trinucleotide context, different trinucleotide contexts varied between the CHH, CHG and CpG contexts (Appendix A). Both the total number and the trinucleotide context number showed similar patterns between the young group (1 year old) and the adult group (9 years old). The methylation levels in nine different regions of the autosome, X chromosome and Y chromosome were further statistically analyzed (Appendix A). The lowest methylation level was found in the 5′UTR, while the highest level was in the gene body and intron region across all chromosomes. Methylation levels on the sex chromosomes showed more variations than the autosome.

### 2.3. DMGs and ASMs from the Thymus between the Two Age Groups

A total of 29,643 DMRs were identified from the thymus between the two age groups (Appendix A). Among them, 1562 DMRs (5.27%) were located in the CpG island (CpGi) (Figure 2A), including 28.62% and 23.11% annotated to the CpGi shore and CpGi shoreshelf, respectively. We annotated 28,607 DMRs (96.50% of 29,643 DMRs) to the gene and the majority of the differentially methylated genes (DMGs) (83.11%) were located in introns (Figure 2B,C, Appendix A). We identified the highest number of DMGs on chromosome 1 and the lowest on chromosome 18 (Figure 2D). The numbers of DMRs were highly correlated with the number of DMGs on the autosome and the X chromosome (R = 0.9160558, *p*-value = 5.6 × 10^−9^, Pearson’s product-moment correlation, 95 percent confidence interval = [0.8012, 0.9658]).

Among the DMRs, 7984 were hypermethylated (hyper) and 21,659 were hypomethylated (hypo) in the adult group compared with the young group. These DMRs were related to 10,666 DMGs, including 1323 hyper, 3702 hypo and 5641 complex methylation (either hyper or hypo) in the young group (Figure 1B). GO and KEGG enrichment analyses of the DMGs found that the hyper DMGs were significantly enriched in 274 GO terms and 30 KEGG pathways, such as cortisol synthesis and secretion (KEGG:04927); oxytocin signaling pathway (KEGG:04921); and parathyroid hormone synthesis, secretion and action (KEGG:04961) (Appendix A). Meanwhile, the hypo DMGs were significantly enriched in 99 GO terms, such as protein binding (GO:0005515), developmental process (GO:0032502) and integral component of postsynaptic density membrane (GO:0099061).

We analyzed the top 2000 DMGs according to the *p*-values due to the limitation of the amount on the STRING website. The protein–protein interaction (PPI) network analysis of DMGs obtained 1042 intersections between the 2000 DMGs. A cytoHubba analysis identified ten hub DMGs, which were *ADCY8*, *PRKCB*, *PTPN11*, *FN1*, *PPP1CC*, *ESR1*, *LOC708441* (*ENSMMUG00000009948*, *HDAC1*), *POLR1B*, *ADCY6* and *PIK3R3* (Figure 2E). These ten hub genes were mainly related to energy metabolism, development process and body protection. Of them, seven hub DMGs decreased methylated levels and three (*LOC708441* (*ENSMMUG00000009948*), *PPP1CC* and *PTPN11*) increased methylated levels in the adult group. MCODE analysis identified 40 MCODE models. Genes of the top eight MCODE models were mainly related to the innate immune system, metabolism and cell cycle (Figure 2F).

Allele-specific DNA methylation (ASM), which is a hallmark of imprinted genes, plays an important role in the growth and development of mammals. We analyzed a genomic landscape of allele-specific DNA methylation in the rhesus macaque thymuses and explored age-related allele-specific DNA methylation. We obtained 293,971,690 SNVs and found 6,381,799 SNVs (2.17%) from the mGAP database [23]. Then, we identified 1992 high-confidence autosomal ASM sites, 367 (18.42%) clustered S-ASM cytosines and 31 allelic differentially methylated regions (aDMRs) on heterozygous loci (Appendix A). The young group had 18 aDMRs and the adult group had 13 aDMRs. Previously, Zhou et al. [24] reported 24,557 allele-specific expressed genes (ASEGs) of 91 human tissues and cell lines of the allele-specific DNA methylation database. Of these 24,557, five genes were shared with our 15 aDMR-related genes in the thymus, which were *U6*, *ZNF287*, *PELO*, *ZNF596* and *NUPR2*.

### 2.4. DEGs of the Thymus between the Two Age Groups in Macaques

The transcriptomes of the thymus for the young and adult groups were analyzed. After RNA-seq data processing, quality control and filtering, we obtained 13.25~19.83 gigabases (GB) in clean bases at an average overall alignment rate of 90.54% (Appendix A). The average number of transcripts per million RNA molecules (TPM) was six (Appendix A). A total of 786 significant differentially expressed genes (DEGs) were identified, including 360 upregulated and 426 downregulated DEGs in the adult group (Figure 3A). The upregulated DEGs in the adult group were enriched in 775 GO terms and 16 KEGG pathways, and downregulated genes were enriched in 570 GO terms and 17 KEGG pathways. Most of these terms and pathways were related to immunity and blood clotting. In particular, genes in the complement system, cytokine (such as interleukin-8 and interleukin-6), and phagocytosis pathways or terms were upregulated in the adult group, while genes in the T cell differentiation and activation, pre-B cell differentiation, cytokine (such as interleukin-4), and thymus development pathways or terms were downregulated (Figure 3B, Appendix A). In addition, the upregulated DEGs in the adult group were enriched in the pathway of blood coagulation and hemostasis, while the downregulated genes were enriched in hemopoiesis.

PPI analysis of DEGs obtained 438 intersections from 786 DEGs and identified ten hub DEGs (*AURKA*, *AURKB*, *BUB1*, *CCNA2*, *CCNB1*, *CDC6*, *CDK1*, *MCM2*, *MCM4* and *PCNA*). These hub DEGs were mainly related to cell-cycle-related enzymes and they were all downregulated in the adult group compared with the young group (Figure 3C). MCODE analysis identified that these models mainly represented DNA replication, the cell cycle, mitosis, the complement system and T cell development (Figure 3D).

### 2.5. Associations of Methylation and Expression

After combining the RNA-seq data and the DNA methylome data, we evaluated the associations between the DNA methylation (DNAm) levels and mRNA expression in four genomic regions (the promoter, gene body, exon and intron regions) (Appendix A). Given that methylation levels in the CHH and CHG contexts were low, very small R coefficients (close to zero) were observed in all regions of the two contexts. However, higher R coefficients were found in the four regions of the CpG context (Figure 4A,B). In the promoter regions, gene expression was negatively correlated with the CpG methylation level (−0.5 < R < −0.01, 2.2 × 10^−16^ < *p* < 0.007), while in the gene body, exon and intron regions, the gene expression was positively correlated with methylation level in both age groups (0.06 < R < 0.22, *p* < 2.2 × 10^−16^).

We identified 431 overlapping genes of DEGs and DMGs between the two age groups (Figure 4C, Appendix A). Among them, mRNA expressions of 198 genes were positively correlated with their methylation level, while the expressions of 233 genes were negatively correlated with the methylation level (Figure 4D). Annotation of the overlapping genes identified 45 immune-related genes, mRNA expressions and methylation levels in the 45 genes between the thymuses of the two age groups (Table 1). These immune-related genes were mainly enriched in the complement and coagulation cascades (KEGG:04610), immune system process (GO:0002376) and system development (GO:0048731). In particular, in the pathway of the complement and coagulation cascades, we identified four genes (*C3*, *C7*, *F13A1* and *CPB2*) with significantly changed gene expressions and methylation levels between the thymuses of the two age groups. Both *C3* and *C7* were important genes in the complement cascade. Compared with the thymuses of the young group (henceforth “young thymus”), the hypomethylated *C3* (intron) and hypermethylated *C7* (exon and intron) both exhibited significantly increased expressions in the thymuses of the adult group (henceforth “adult thymus”), leading to the activation of a complement system in the adult thymus. Meanwhile, both *F13A1* and *CPB2* were important genes in coagulation cascades. *F13A1* exhibited hypomethylation (intron) and significantly higher expression in the adult thymus, while *CPB2* exhibited hypomethylation (intron) and lower expression. Other blood-coagulation-related genes were also found in the overlapping genes, such as *CLEC4D*, *CLEC4E*, *FCN3*, *FGF2* and *FGF7*. All of them exhibited hypomethylation and significantly higher expression in the adult thymus. These indicated that blood coagulation was enhanced in the adult thymus. In addition, genes related to T cell differentiation, selection and maturation, including *CD3G*, *GAD2* and *ADAMDEC1*, were found in the overlapping genes, and all exhibited hypomethylation and lower expression in the adult thymus compared with the young thymus. Decreased expression of these genes may be associated with a decrease in the function of T cell differentiation, selection and maturation in the adult thymus (Figure 4E).

### 2.6. Promoter Methylation

Local CpG methylation (mCpG) levels in a promoter are known to be associated with gene expression. A total of 2340 promoters were shared by the four individuals (Figure 5A). The shared promoters included 295,957 loci (about 75%) of the CHH context, 81,355 loci (about 20%) of the CHG context and 15,868 (about 5%) loci of the CpG context. Compared with CHH or CHG, CpG in promoters showed a relatively higher level of average methylation (0.8–0.9) (Figure 5B). In addition, we identified 37 unmethylated promoters, 557 heterogeneously methylated promoters and 678 hypermethylated promoters across all four thymuses of the macaques (Appendix A). Compared with the young thymus, the adult thymus showed a slightly decreased average methylation level of CpG promoters and we identified seven promoters that significantly changed methylation levels between the two groups (Appendix A). The promoters of *ENSMMUG00000042906* and *mml-mir-155* showed significantly increased methylation levels and the promoters of *MAN1A1*, *ENSMMUG00000013570*, *ENSMMUG00000048040*, *ENSMMUG00000041194* and *GJA10* showed significantly decreased methylation levels in the adult thymus. In addition, we obtained 823 potential promoter regions of gender–age-related genes based on the methylation level trends (Appendix A). The *DCLK3* promoter and *OR10G8* promoter may be the negative gender–age-related regions.

### 2.7. Comparisons of the Age-Related Methylated Genes in Four Mammals

To investigate whether the age-related methylated genes were conserved among mammal species, we downloaded previously reported methylation data from 37 tissues/cell lines of four species (including 5978 samples) (Appendix A). Combined with 431 age-related methylated genes in the macaque thymus, we obtained a total of 3965 age-related homologous genes (Figure 6). The numbers of age-related methylated genes shared with species were much less than those of species-specific genes (351–1785). Only two age-related methylated genes (*CXXC5* and *TBX5*) were shared in the four species. There were 34 age-related methylated genes unique to humans and rhesus macaque, 24 of which were found in the thymus of the rhesus macaque, indicating they were probably conserved age-related methylated genes in primates.

## 3. Discussion

### 3.1. DNA Methylation Profiles of the Thymus in the Rhesus Macaque

As one of the essential immune organs, the thymus is where T lymphocytes develop, differentiate and mature, and the thymus plays a vital role in the development of peripheral lymphocytes. The thymus is different from other tissues in its development, where it reaches its maximum size in infancy and gradually degenerates after puberty, eventually being replaced by adipose tissue [25,26]. Previous studies about the development and aging of the thymus mainly focused on the associated morphology, histology, endocrinology and transcriptomics [27,28,29]. However, the landscape of DNA methylation in the thymus and its dynamic changes with age are still poorly understood, especially in the non-human primate rhesus macaques. This study obtained 447.56 GB of high-quality DNA methylation data from the thymuses of four rhesus macaque individuals. The unique alignment rate of DNA methylation data was more than 69%. The DNA methylation landscape in the rhesus macaque thymus was characterized. Thymus tissue showed an average methylation level (10 kB/bin) of 0.2, both CHH and CHG had a methylation level close to 0, and CpG had a methylation level of approximately 0.7. Methylation levels of different genomic regions indicated that the gene body had the highest methylation level and the 5′UTR regions had the lowest. Most of the characteristics of the thymus methylome were similar to other tissues, such as the cortical development, liver and pancreas of giant pandas, and the blood of rhesus macaques [30,31,32]. To investigate whether the individual variations had biased our results, PCA and Spearman’s correlation analysis were conducted. The results indicated that the methylation level was basically determined by the genomic regions; however, within each genomic region, age also contributed to the methylation level of all autosomal genes. Future studies are needed that involve more thymus samples from individuals of varied ages to comprehensively reveal methylation profiles of the thymus in macaques.

### 3.2. Age-Related Alterations of DNA Methylation and Gene Expression

Previous studies on the thymus of humans, chickens and mice indicated that the significant physiological changes in the thymus with age were the activation of the complement system, the enhancement of blood clotting and the changes in endocrine hormones [33,34,35]. We compared two young thymuses and two adult thymuses and identified 786 DEGs in the transcriptome analysis and 10,666 DMGs in the methylome analysis. DMGs enrichment and hub DMGs demonstrated that these genes were mostly related to energy metabolism, innate immunity, and hormone synthesis and secretion, such as cortisol and parathyroid hormone. Meanwhile, DEGs and hub DEGs identified based on transcriptome analysis were mostly enriched in immune (mainly complement system), clotting and other pathways. Our results implied that the thymus in adult macaques significantly changed in immune responsiveness, coagulation and hormonal endocrine activity compared with young macaques, which was consistent with results from previous physiological studies [33,34,35]. Combining data from DNA methylation and transcriptomes, we identified 431 overlapping genes of DMGs and DEGs, where 45 genes were related to immune functions. Our results indicated that when the thymus aged, the expression of these genes, including immune-related genes, changed significantly and the process might have resulted from the changing methylation levels of these genes.

It is well known that a promoter is a crucial genomic element in regulating gene expression and its methylation level is associated with the repression of transcription [21,36]. We identified 2304 shared promoters in the thymus and of these, 37 were unmethylated in all samples, indicating that they may be housekeeping genes, where expression was irrelevant to DNA methylation [37]. In addition, 678 promoters were universally hypermethylated in the thymus, suggesting they were genes with low expression in the thymus. We also identified seven promoters with significantly changed methylation levels between the young and adult thymuses. Horvath et al., found four CpG sites in the *KLF14* promoter that were important epigenetic clocks in multiple tissues (without thymus) of macaques [38]. However, these sites in the *KLF14* promoter did not show age-related methylation in the thymus in our study. Combining DNA methylome and transcriptome data allowed us to examine the correlation between DNA methylation and mRNA expression in the promoter regions. It was found that the CpG methylation level in the promoter region was negatively correlated with the mRNA expression level, while in the gene body, it was positively correlated with the mRNA expression level. The results were consistent with methylome studies on the spleens of Meishan piglets and the brains of rhesus macaques, suggesting the expression of the genes controlled by them was repressed [39,40]. Except for the promoter regions, the association between the methylation level and mRNA expression in other genomic regions appears complex. Either hyper- or hypomethylation in these regions can lead to promoting/repressing gene expression. In particular, in the 431 overlapping genes, the number of genes with methylation levels positively correlated with mRNA expression was approximately equal to the number of genes with negative correlation. Consequently, the mechanism of how DNA methylation affects gene expression in these regions is worthy of further investigation.

### 3.3. Changes in Immune Function of the Thymus with Age

Given its importance as an immunity-related organ, combining DNA methylation and transcriptome analyses of young and adult thymuses allowed us to investigate how the expressions of immune-related genes changed with age and how DNA methylation affected the expressions of immune-related genes. We found from transcriptome data that DEGs were enriched in pathways of the complement system and coagulation cascade, as well as T cell development, differentiation and maturation. Moreover, several important complement genes, such as *C3*, *C7* and *C5* receptor 2 (*C5AR2*), were found in the DMGs and DEGs overlapping genes. Both *C3* and *C7* are key genes in the complement system pathway (KEGG:04610). The C3 complement can be cleaved into C3a and C3b by enzymes. Anaphylatoxin C3a binds to its receptor (C3aR) to recruit leukocytes, and thus, activate the classical pathway of the complement system. Meanwhile, C3b binding to C5 also promotes the formation of complement complexes, such as C6, C7, C8 and C9, through a series of chain reactions. These complement complexes can bind the target cell membrane and form pores for promoting the cell rupture of pathogenic microorganisms [35,41,42]. Our results showed that in the adult thymus, *C3* represented hypomethylation with a significantly high expression, indicating an activation of the classic complement system leading to an enhancement of immune function in the adult rhesus macaques. In addition, the *C7* gene in adult macaques showed hypermethylation and upregulated expression, while the *C5AR2* gene was hypomethylated and upregulated expression compared with young macaques. Changes in both *C7* and *C5AR2* can activate the alternative complement system. Our results demonstrated that the hypomethylation of *C3* and *C5AR2* genes plus the hypermethylation of the *C7* gene led to the upregulation of expression in these key genes, eventually activating both the classic and the alternative complement system and enhancing the innate immune function in the adult rhesus macaque. We concluded that DNA methylation is an important regulatory mechanism involving age-related immune function changes in the rhesus macaque thymus.

In humans and other mammals, the coagulation potential gradually increases with age [19,34,43]. The increase in blood coagulation potential with age is mainly due to the increased concentrations of most procoagulant factors in plasma [34,43]. The ability of rapid coagulation is also considered to be an immune mechanism against pathogens [44]. Young and adult thymus transcriptomes were found that DEGs were enriched in coagulation cascades. We also identified eight DMG and DEG overlapping genes associated with the coagulation cascade, which were *F13A1*, *CPB2*, *CLEC4D*, *CLEC4E*, *FCN3*, *PDGFRA*, *FGF2* and *FGF7*. Except for *CPB2*, the other seven genes showed hypomethylation and significantly upregulated expression in the two adult thymuses compared with the two young thymuses. *F13A1* encodes the coagulation factor XIIIA1 peptide, which is the last activated proenzyme in the coagulation cascade. The upregulated expression of *F13A1* can promote the formation of cross-linked fibrin between the fibrin monomer and the soluble fibrin polymer, and eventually generates fibrin clotting [45,46]. The expression of *FCN3* increased in the adult thymus compared with the young thymus. *CLEC4D* and *CLEC4E* encode c-type lectin and c-type lectin-like domains, respectively. *PDGFRA* encodes the platelet-derived growth factor receptor and *FGF2* and *FGF7* are fibroblast growth factors. All the genes encode factors that directly participate in blood coagulation. The upregulated expressions of these genes implied stronger blood-clotting abilities in adults than in young rhesus macaques. In contrast, the carboxypeptidase gene (*CPB2*) was hypomethylated and showed downregulated expression in the adult macaques. It led to a reduction in the fibrin clotting degradation in the fibrinolytic system and inhibition of fibrin clotting degradation. Our study demonstrated that the coagulation potential was enhanced with increasing age in two ways. First, the hypomethylation and upregulated expression of procoagulant genes increased the concentrations of procoagulant factors in the blood. Second, the hypomethylation and downregulated expression of *CPB2* reduced the fibrin clot degradation. The enhancement of coagulation with age is potentially beneficial by reducing the risk of bleeding, promoting wound healing and resisting the invasion of pathogenic microorganisms [45,46].

The thymus tissue gradually degenerates and immune function declines with increasing age [5]. Atrophy and degeneration of the thymus reduce the output of naïve T cells and changes T cell phenotypes and the peripheral T cell composition [47]. Our results showed that both DEGs and the DMG and DEG overlapping genes included genes related to T cell differentiation, selection and maturation, implying T cell functional changes in aging thymuses. We identified four genes related to T cell differentiation, activation and selection (*CD3G*, *GAD2*, *ADAMDEC1*, *LCK*) and a thymogenic hormone gene (*TMPO*). Both the methylation level and mRNA expression of the five genes changed significantly with age. *CD3G*, *GAD2* and *ADAMDEC1* showed hypomethylation and downregulated expression in the adult rhesus macaque thymus. The downregulated expression of T cell surface glycoprotein CD3 gamma chain gene (*CD3G*) decreases T cell phenotypes that reduce the ability of T cells to recognize pathogens [5]. Glutamic acid decarboxylase 2 (*GAD2*) is the target gene of self-reactive T cells, where downregulation of its expression causes a decrease in immune responses [48]. Meanwhile, the downregulated expression of the ADAM-like decysin 1 (*ADAMDEC1*) gene can reduce the function of dendritic cells and weaken their interaction with germinal center T cells [49]. In contrast, the leukocyte C-terminal Src kinase gene (*LCK*) that encoded a key signaling molecule in T cells selection and maturation showed hypermethylation and downregulated expression in the adult macaque thymus. The downregulated expression of *LCK* could reduce T cell immunity, suggesting the degeneration of the thymus. Moreover, we found the thymogenic hormone gene (*TMPO*) had hypomethylation and downregulated expression in the adult thymus. The downregulated expression of *TMPO* could inhibit immune cell maturation [50]. Therefore, due to changes in the methylation level of *TMPO* and several genes related to T cell differentiation, activation and selection, the expression of these genes will down-regulate and may have a great contribution to the decrease in immune function in T cells and the gradual degradation of the thymus in aging macaques.

### 3.4. Comparisons of Age-Related Methylated Genes in Four Mammals

Previously, many age–methylation clocks were reported based on multiple tissues of multiple species [38,51,52,53,54,55,56,57,58,59], but these studies did not include the thymus. When combining the reported data and the 431 thymus age-related methylated genes in the present study, we identified 3965 age-related homologous genes in humans, rhesus macaques, mice and dogs. However, most of these genes were species-specific and only two of them were shared by all four species, implying that age-related methylated genes varied significantly between species. Importantly, we identified 34 age-related methylated genes that were probably conserved in primates, 24 of which were identified in the thymus in the present study. In particular, Horvath et al. [38] reported a total of 255 key age–methylation clock CpG sites based on rhesus monkeys’ multi-tissues (without thymus). We then checked the 255 clock CpG sites in our dataset of thymus and found that 13 CpG sites in five genes (*ARID5B*, *TBR1*, *KLF4*, *ZNF507* and *SMG6*) were also age–methylation clock CpG sites of the thymus (Appendix A). These results have largely extended our understanding of age–methylation clocks in primates.

## 4. Materials and Methods

### 4.1. Sample Collection

Sample collection and utility protocols were approved by the Sichuan Hengshu Bio-Technology Co., Ltd. (Chengdu, Sichuan Province, China) and were operated by a professional veterinarian. Animal procedures were approved by the Institutional Animals Care and Use Committee, and the research was approved by the Ethics and Animal Welfare Committee of Sichuan University (no. SCU220429001). Our experimental procedures complied with the current laws on animal welfare and research in China. Homogenate postmortem thymus tissues were collected from two 1-year-old rhesus macaques (one female, one male) and two 9-year-old rhesus macaques (one female, one male) (Appendix A). The samples were immediately stored in liquid nitrogen and RNAlater (Ambion Inc., Austin, TX, USA). Total RNA was extracted using TRIzol reagent (Invitrogen, Carlsbad, CA, USA) following the manufacturer’s protocol and treated with RNase-free DNase I [60].

### 4.2. Whole-Genome Bisulfite Library and RNA Library Preparation and Sequencing

Genomic DNA extraction was performed using the DNeasy Blood and Tissue Kit (Qiagen, Hilden, Germany). Bisulfite conversion of 600 ng genomic DNA was performed with the EZ DNA methylation kit (Zymo Research, Irvine, CA, USA). Sequencing libraries were made with Illumina TruSeq DNA Methylation library preparation kits. A lambda DNA sequence was spiked in at 1% concentration to assess the bisulfite conversion efficiency. Libraries were spiked with 10% PhiX to improve the base calibration calls and subsequently sequenced on an Illumina X-Ten Platform with paired-end (PE) reads (2 × 150 bp). All reads of DNA methylation sequence were submitted to the NCBI Sequence Read Archive with SRA numbers SRR20950188–SRR20950191.

Samples were required to achieve an RNA integrity number (RIN) of at least 7.0, an OD260/280 value of 2.0 and an RNA concentration of at least 588 ng/μL using a NanoDrop2000 micro-spectrophotometer, Agilent RNA 6000 Nano Kit and Agilent 2100 Bioanalyzer (Agilent, Waldbroon, Germany) (Appendix A). RNA library construction and sequencing were performed using the same protocols employed by a previous study [33]. All reads of the RNA sequence were unpublished data of the key laboratory of bio-resources and eco-environment of the Ministry of Education, College of Life Sciences, Sichuan University, China.

### 4.3. Data Quality Control and Alignment

We trimmed the low-quality and adapter-containing portions of the reads using Trim Galore! v0.5.0 with the default parameters. We aligned the clean data to the bisulfite-treated *Macaca mulatta* reference genome (NCBI Assembly ID GCA_000772875.3) using Bowtie2 v2.3.4.1 [61] with -N 0 -L 20 -non_directional -p 3 -multicore 3, with the other parameters set as the defaults, which we created using the Bismark v0.20.0 [62] bismark_genome_preparation program. Alignments with evidence of duplication were removed using the deduplicate_bismark program of Bismark v0.20.0 with the -paired option. The Bismark v0.20.0 [62] bismark_methylation_extractor program was run on one deduplicated BAM file per sample to extract DNAm levels in the CHH, CHG and CpG contexts with --no_overlap --ignore_r2 2 --ignore_3 prime_r2 2, with the other parameters set as the defaults.

The adapter and low-quality reads were trimmed by using Trim Galore! V0.5.0 with the default parameters (http://www.bioinformatics.babraham.ac.uk/projects/trim_galore/ (accessed on 10 July 2019)). We aligned the clean reads for each sample to the MMUL_8.0.1 *Macaca mulatta* reference genome (NCBI Assembly ID GCA_000772875.3, Ensembl release 92 annotation), which we created using the HISAT2-stringtie-DESeq2 program [63]. Clean reads were mapped to the reference genome with the splice-aware aligner HISAT2 v2.1.0 [64]. The resulting alignment (SAM) files were sorted and converted using samtools v1.6 to produce one BAM file per sample [65]. Using StringTie v2.0.4 [66] and Ballgown [67], we processed the counts and the relative transcriptional level of unigene was calculated using the TPM method for transcripts per million [68]. Then, we filtered out all cases where the counts of genes were less than five. Differentially expressed genes were determined with |log_2_ fold change|> 1 and adjusted *p*-value < 0.05 with the Benjamini–Hochberg FDR method using DESeq2 [69].

### 4.4. Differentially Methylated Regions Analysis and Annotation

The PCA was performed on the WGBS data using the R pheatmap package (https://www.R-project.org/ (accessed on 6 October 2022)). We calculated the Spearman correlation of each genomic region for four samples and plotted the results.

The detection of differentially methylated regions (DMRs) in the CpG context of 5 mC between two ages was identified by using the DMAP [70] diffmeth program with -z -A 40,220 -W 1000 -U 0.005 -N -I 2, and the other parameters set as the defaults. DMRs were annotated to differentially methylated genes by using the DMAP [70] identgeneloc program with -d 2000 -t -U -I -I -K -R, and other parameters were set as the defaults. The DMAP identgeneloc program found that the CpG island shore was located as far as 2 kb from the CpG islands and the CpG island shelf was between 2 kb and 4 kb away from the CpG islands. The DMRs between two ages were filtered with a *p*-value < 0.05, CpG counts per 1000 nt window ≥ 5 and sample counts per group = 2. These DMRs were annotated to the DMGs. Gene Ontology and KEGG (Kyoto Encyclopedia of Genes and Genomes) enrichment analyses of the DEGs and DMGs were performed with the default parameters using g:Profiler, respectively [71]. Significance was considered as a corrected *p*-value < 0.05 with the Benjamini–Hochberg FDR method.

### 4.5. Promoter Methylation Analysis

Promoters in the CpG context of 5 mC are key targets for epigenetic modulation but their locations remain unknown for *Macaca mulatta*. A promoter was defined as being 1000 bp upstream from gene transcript start sites (TSS) [72], and the 3′UTR, 5′UTR, CDS and intron data were downloaded from the *Macaca mulatta* genome annotation (Ensembl release 92). According to the extent of the methylated level, the promoters were classified as (1) unmethylated promoters (0, 0.2), (2) heterogeneously methylated promoters (0.2, 0.8) and (3) hypermethylated promoters (0.8, 1) [73]. The difference in methylation level of promoters between the two groups was greater than two-fold, and they had a *p*-value < 0.05 using a chi-squared test [72].

### 4.6. Associations of Methylation and Expression

With WGBS data and RNA-seq data, we computed the average gene expression per group and correlated these values to average DNAm levels in the same groups in both the CHH, CHG and CpG contexts at the promoter and the gene body, exon and intron regions. With the TPM value and methylation of four regions per gene, we used the R ggplot2 package [74]. To assess whether age confounded the relationship between methylation and expression, we used a multiple linear regression model that adjusted for age and checked whether the methylation coefficient was FDR < 5% with Pearson’s product-moment correlation method.

We also analyzed the shared genes in both promoter-methylated genes and DEGs. We calculated the correlation for the methylation levels for the CpG context and TPM with the default parameters using the cor.test function of the R package stats (https://www.R-project.org/ (accessed on 7 June 2020)). Genes where the methylated level of promoter-methylated genes decreased but the expression of DEGs increased were called negatively correlated genes. Genes where the methylated level of promoter-methylated genes increased but the expression of DEGs increased were called positively correlated genes.

### 4.7. Protein–Protein Interaction Network Analysis

A protein–protein interaction network analysis was performed on the DMGs (highest confidence of minimum required interaction score: 0.900) and DEGs (high confidence of minimum required interaction score: 0.700) using the STRING web server (https://string-db.org (accessed on 20 July 2020)) v11.0 [75]. The network plot was drawn using Cytoscape v3.7.1 [76]. The intersections of genes were calculated using eight models with the default parameters with a multi-protein complexes clustering algorithm using the Molecular Complex Detection (MCODE) [77] plugin of Cytoscape v3.7.1 [76]. PPI analysis of the DEGs showed that the 10 hub genes were identified with 12 algorithms—Matrix Clique (MCC), Density of Maximum Neighborhood Component (DMNC), Maximum Neighborhood Component (MNC), Degree, Edge Percolated Component (EPC), BottleNeck, EcCentricity, Closeness, Radiality, Betweenness, Stress, ClusteringCoefficient—using the cytoHubba plugin of Cytoscape v3.7.1 [76,78].

### 4.8. Gene Annotation and Multi-Species Synteny

We downloaded about 11,700 age–methylation clock CpG sites and 137 age-related DMRs from multiple mammals, including humans [51,53,79,80,81], rhesus macaques [32], mice [54,55,57,59] and dogs [58]. These CpG sites and DMRs were mapped to the genomes of the corresponding species (human: GRCH38.p13; rhesus macaque: Mmul_10; mouse: GRCm39; dog: ROS_Cfam_1.0) using the Ensembl BioMart web server (Ensemble Genes 107) [82,83]. A total of 431 age-related genes in our study were mapped to the macaque genome (Mmul_10). Then, all genes were mapped to the human genome (GRCH38.p13) where possible so that functional analysis tools with access to the most detailed annotations could be utilized.

### 4.9. Identification of ASM Sites and aDMRs

Deduplicated BAM files were sorted, indexed using samtools v1.9 [65] and transferred into the ATCGmap format file using the CGmapTools v0.1.2 [84] cgmaptools_convert _bam2 cgmap program with the --rmOverlap option. Then, the heterozygous SNV calls and VCF files per sample were predicted using the CGmapTools v0.1.2 cgmaptools_snv program with the --bayes-dynamicP option before filtering with >1 × 10^−7^ as the SNV site’s *p*-value and inexplicit prediction sites using the Python script1. The allele-specific DNA methylation analysis was performed using the CGmapTools v0.1.2 cgmaptools asm with the -r option. The ASM was determined for each of the cytosines that were covered by at least 15 reads on each allele. Merged genomic regions that had five or more allele-specific methylated cytosines were reported as aDMR [72]. The S-ASM cytosines in aDMR were called clustered S-ASM. Merge aDMRs with a distance of less than 10 bp were reported as a large fragment of aDMR and the length was calculated with the Python script2.

Python and R scripts are available at Github (https://github.com/Grace0114/DNA-Methylation-Analysis-of-Rhesus-Macaque-Thymuses (accessed on 20 July 2022)).

## 5. Conclusions

In conclusion, we characterized the DNA methylation profiles of two young and two adult rhesus macaque thymuses and identified 786 DEGs and 10,666 DMGs. After combining the two-omics data, we found dynamic changes in DNA methylation and gene expression, immunity function changes regulated by DNA methylation, and specific ASMs between the two young and two adult rhesus macaque thymuses. We suggested that DNA methylation contributed greatly to age-related immune function changes in the thymus, particularly involving the activation of the complement system, enhancement of coagulation, and the suppression of T cell development, differentiation and maturation. Our findings provided a better understanding of epigenetic changes in thymus senescence. Given our small sample data, we need large-scale samples to study the molecular mechanism of thymus senescence in the future.

## Figures and Tables

**Figure 1 ijms-23-14984-f001:**
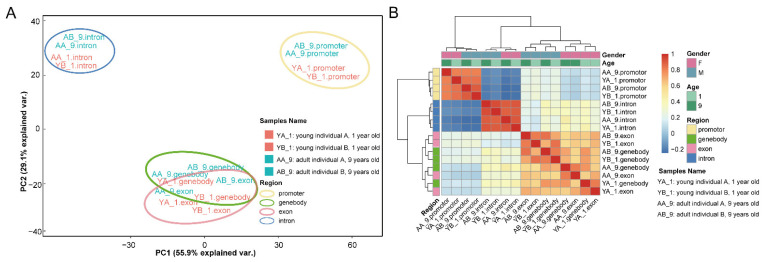
PCA and Spearman’s correlation analyses of four individuals based on the DNA methylation levels of the genomic regions (promoter, gene body, exon and intron). (**A**) PCA analysis of 29,324 autosomal genes. (**B**) Heatmap of Spearman’s correlation analyses of 29,324 autosomal genes.

**Figure 2 ijms-23-14984-f002:**
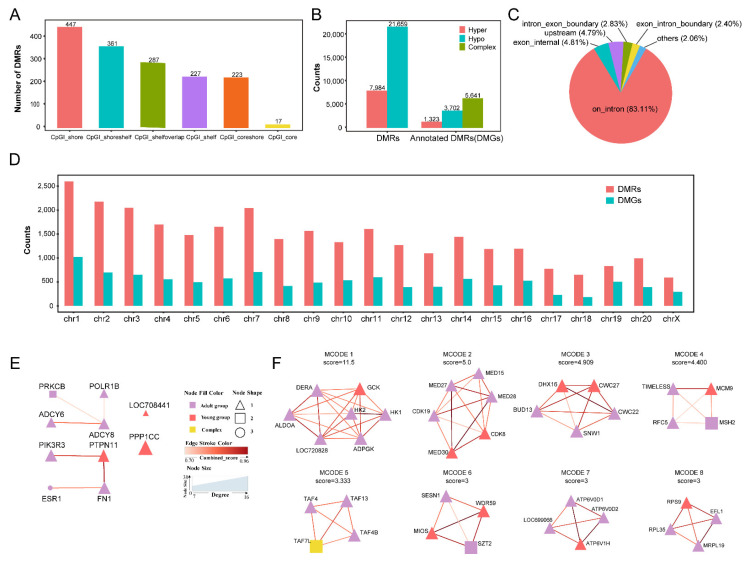
Differentially methylated regions in the adult group compared with the young group. (**A**) A total of 1562 differentially methylated regions (DMRs) (5.27% of 29,642 DMRs) were annotated to the CpG island location. (**B**) The number of hypercomplex DMRs and annotated DMRs (DMGs). (**C**) Pie chart showing the location of DMRs in the genome. A total of 28,607 DMRs (96.50%) were annotated to the gene location. (**D**) A total of 28,607 DMRs annotated to 10,666 DMGs were distributed on the chromosomes. The number of DMRs was highly correlated with the number of DMGs (R = 0.9160558, *p*-value = 5.6 × 10^−9^, Pearson’s product-moment correlation, 95 percent confidence interval = [0.8012, 0.9658]). (**E**) The network of ten hub genes with 12 algorithms using the cytoHubba plugin of Cytoscape v3.7.1. The color of each node represents the change in the DMR methylated level in the adult group compared with the young group (red, upregulated; purple, downregulated; and yellow, complex). The color gradation of the edge stroke is representative of the combined sore between the two genes. The node size represents the degree level of the hub genes. The node shape represents how many DMG had DMRs (triangle, only one; square, two; and circle, three). (**F**) The networks of the top eight MCODE models with a multi-protein complexes clustering algorithm using the Molecular Complex Detection (MCODE) plugin of Cytoscape v3.7.1. The color gradation of the node is representative of the log_2_ fold change. The color of the edge stroke represents the combined sore between two genes. The node size has no meaning.

**Figure 3 ijms-23-14984-f003:**
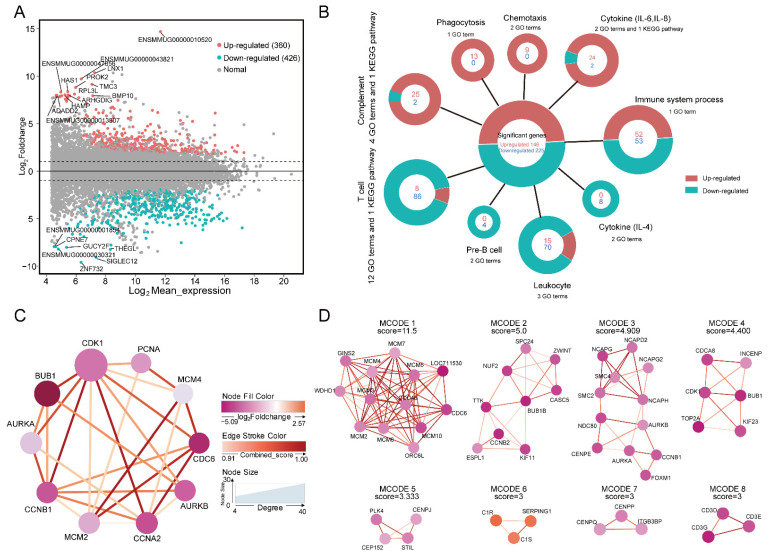
Differentially expressed genes (DEGs) in the adult group compared with the young group. (**A**) MA plot showing the distribution of the log_2_ mean expression plotted against the log_2_ fold change for each gene. Upregulated and downregulated genes are depicted by red and blue dots, respectively (n = 2 per group). Grey dots indicate non-differentially expressed genes. The top 20 most differentially expressed genes are labeled. The numbers of up- and downregulated DEGs were 360 and 426, respectively. (**B**) GO terms and KEGG pathways enrichment analysis were largely related to immunity in DEGs. A circular diagram showing the upregulated and downregulated pathway terms related to immunity (brown, upregulated; green, downregulated) in DEGs (FDR < 0.05). (**C**) PPI analysis of 786 DEGs in the adult group compared with the young group. The network of ten hub genes with 12 algorithms using the cytoHubba plugin of Cytoscape v3.7.1. The color gradation of the node is representative of the log_2_ fold change. The color of the edge stroke represents the combined sore between two genes. The node size represents the degree level of hub genes. (**D**) The networks of eight MCODE models with a multi-protein complexes clustering algorithm using the Molecular Complex Detection (MCODE) plugin of Cytoscape v3.7.1. The color gradation of the node is representative of the log_2_ fold change. The color of the edge stroke represents the combined sore between two genes. The node size has no meaning.

**Figure 4 ijms-23-14984-f004:**
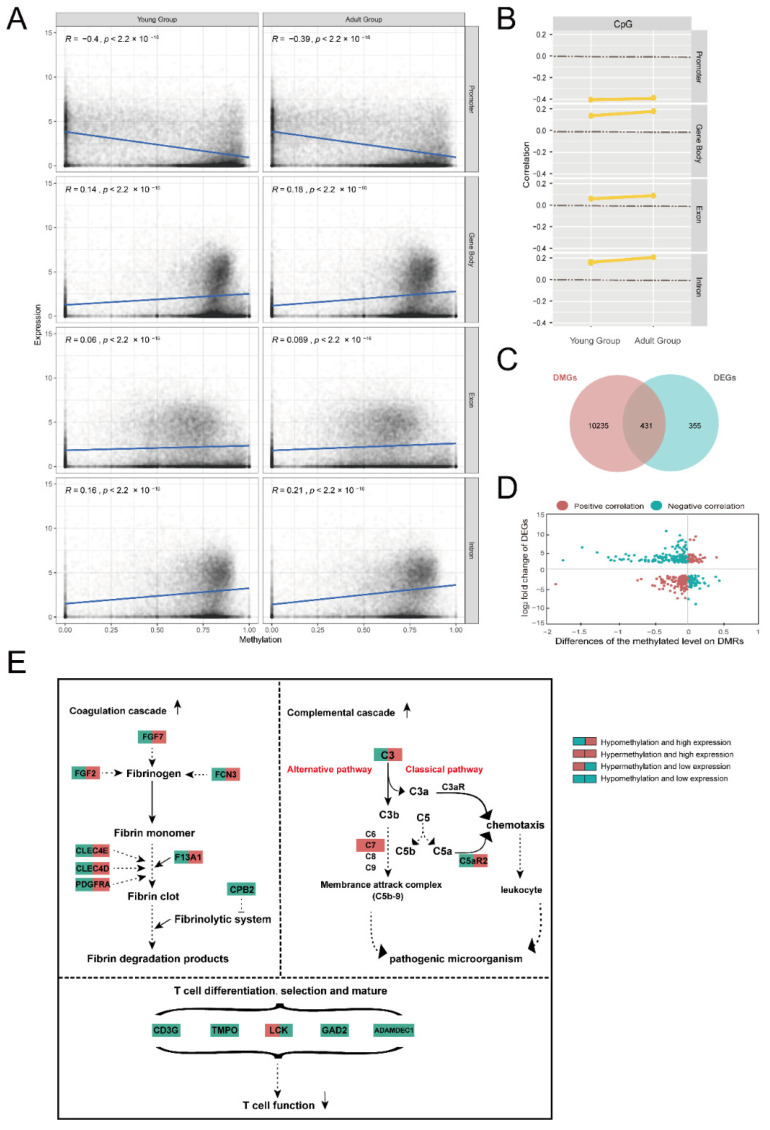
Relationship between methylation and expression. (**A**) Dot chart showing the methylation pairs stratified by age in columns and feature type in rows with 30,807 genes. Each dot represents the mean methylation level of the CpGs within the feature across all samples of that age on the x-axis and the log_2_([mean TPM] + 1) for the promoters, gene bodies, exons and introns on the y-axis. Linear regression with shaded standard error and the rho for each correlation is listed for each plot with Pearson’s method. (**B**) Correlation of feature expression and methylation stratified by cytosine context (columns) and feature type (rows). (**C**) Venn diagram depicting the DEGs, DMGs and shared genes. (**D**) Quadrant diagram showing the distribution of DMGs and DEGs (red (positive correlation): hypermethylation with a high expression level and hypomethylation with low expressions level; green regions (negative correlation): hypermethylation with a low expression level and hypomethylation with a high expression level). (**E**) KEGG and GO enrichment map of the complement system; coagulation cascade; and T cells differentiation, activation and maturation by DEGs and DMGs in the thymus. Green in the left of the box and red in the right of the box: hypomethylation and high expression; red in the whole box: hypermethylation and high expression; red in the left of the box and green in the right of the box: hypermethylation and low expression; green in the whole box: hypomethylation and low expression in the thymuses of the adult group compared with the young group.

**Figure 5 ijms-23-14984-f005:**
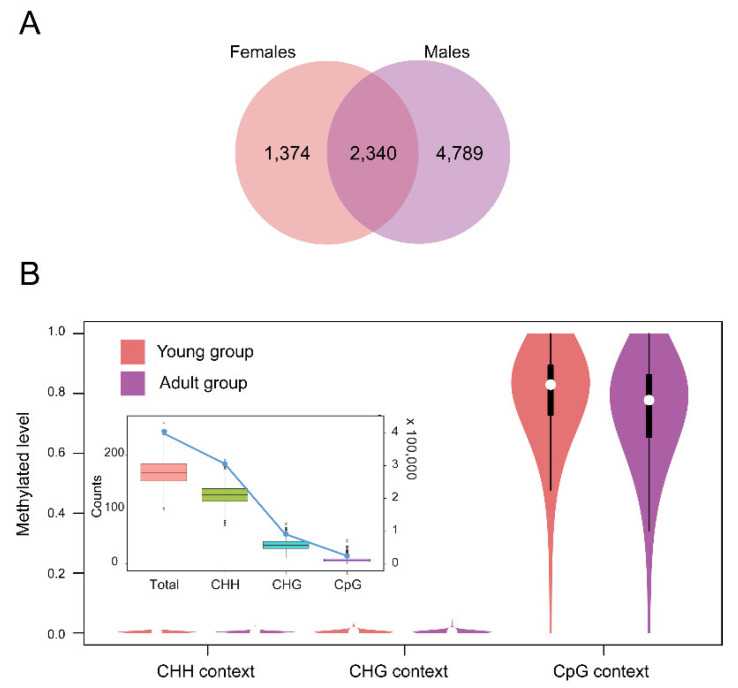
Distribution of promoter methylation in two groups. (**A**) Venn diagram depicting promoter regions and shared promoter regions based on gender. (**B**) Boxplot depicting the counts of the CHH context, CHG context and CpG context according to the reference genome. Vioplot chart depicting the CHH context, CHG context and CpG context of 2340 shared promoters’ distribution across the two groups.

**Figure 6 ijms-23-14984-f006:**
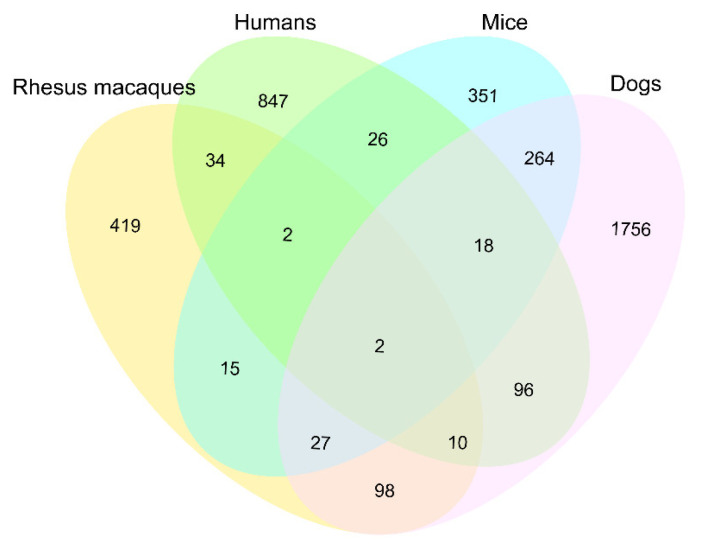
Venn diagram of age-related methylated genes in four mammals (rhesus macaques, humans, mice and dogs).

**Table 1 ijms-23-14984-t001:** Alterations of the methylation level and expression level in the 45 overlapping genes of DMGs and DEGs related to immunity.

Pathway and Term	Pathway and Term ID	Gene Name	Ensembl ID	DNA Methylation Difference	Expression Change
Complement and coagulation cascades	KEGG:04610	*C3*	ENSMMUG00000008693	−	+
Complement and coagulation cascades	KEGG:04610	*F13A1*	ENSMMUG00000001248	−	+
Complement and coagulation cascades	KEGG:04610	*CPB2*	ENSMMUG00000017029	−	−
Complement and coagulation cascades; immune system process	KEGG:04610; GO:0002376	*C7*	ENSMMUG00000014171	+	+
Immune system process	GO:0002376	*C5AR2*	ENSMMUG00000009277	−	+
Immune system process	GO:0002376	*CLEC4D*	ENSMMUG00000013703	−	+
Immune system process	GO:0002376	*CLEC4E*	ENSMMUG00000013706	−	+
Immune system process	GO:0002376	*DNASE1L3*	ENSMMUG00000011235	−	−
Immune system process	GO:0002376	*FCN3*	ENSMMUG00000018322	−	+
Immune system process	GO:0002376	*HELLS*	ENSMMUG00000017255	+	−
Immune system process	GO:0002376	*LCK*	ENSMMUG00000040694	+	−
Immune system process; system development	GO:0002376; GO:0048733	*FANCD2*	ENSMMUG00000008966	−	−
Immune system process; system development	GO:0002376; GO:0048732	*LILRB4*	ENSMMUG00000047009	+	−
Immune system process; system development	GO:0002376; GO:0048734	*PDGFRA*	ENSMMUG00000017395	−	+
Immune system process; system development	GO:0002376; GO:0048731	*RARA*	ENSMMUG00000012486	−	+
System development	GO:0048731	*ADGRG3*	ENSMMUG00000015690	−	+
System development	GO:0048731	*CD3G*	ENSMMUG00000017600	−	−
System development	GO:0048731	*NOX4*	ENSMMUG00000011116	−	+
System development	GO:0048731	*PROX1*	ENSMMUG00000011914	−	+
Carboxy-lyase activity	GO:0016831	*GAD2*	ENSMMUG00000012233	−	−
Cation binding	GO:0043169	*TRAIP*	ENSMMUG00000016476	+	−
Cell division	GO:0051301	*MIS18BP1*	ENSMMUG00000016515	+	−
Cell–cell junction	GO:0005911	*CLDN11*	ENSMMUG00000009274	−	+
Cellular response to stress	GO:0033554	*UBE2T*	ENSMMUG00000013795	−	−
Cytokinesis	GO:0000910	*KIF23*	ENSMMUG00000014887	+	−
DNA polymerase binding	GO:0070182	*FANCI*	ENSMMUG00000011155	−	−
Hydrolase activity, hydrolyzing N-glycosyl compounds	GO:0016799	*NEIL3*	ENSMMUG00000007394	+	−
Hydrolase activity, hydrolyzing O-glycosyl compounds	GO:0004553	*MGAM*	ENSMMUG00000016273	−	+
Immune response	GO:0006955	*CTSV*	ENSMMUG00000022971	−	−
Intrinsic component of membrane	GO:0031224	*TMPO*	ENSMMUG00000023719	−	−
Metallopeptidase activity	GO:0008237	*ADAMDEC1*	ENSMMUG00000005318	−	−
Metallopeptidase activity	GO:0008237	*ADAM28*	ENSMMUG00000005317	−	−
Microfilament motor activity	GO:0000146	*MYH8*	ENSMMUG00000009763	−	−
Molecular function	GO:0003674	*DLEU7*	ENSMMUG00000021744	−	−
Nucleic acid metabolic process	GO:0090304	*MCM4*	ENSMMUG00000015360	+	−
Nucleobase-containing compound metabolic process	GO:0006139	*POLE2*	ENSMMUG00000003913	−	−
Phosphoric ester hydrolase activity	GO:0042578	*PTPN7*	ENSMMUG00000013789	−	−
Plus-end-directed microtubule motor activity	GO:0008574	*KIF11*	ENSMMUG00000023266	+	−
Positive regulation of epithelial cell proliferation	GO:0050679	*FGF7*	ENSMMUG00000009842	−	+
Protein catabolic process	GO:0030163	*LRR1*	ENSMMUG00000037526	+	−
Protein catabolic process	GO:0030163	*LNX1*	ENSMMUG00000041981	−	+
Protein modification process	GO:0036211	*FBXO32*	ENSMMUG00000023778	−	+
Protein-containing complex	GO:0032991	*POLE*	ENSMMUG00000015463	−	−
Regulation of gene expression	GO:0010468	*FGF2*	ENSMMUG00000007419	−	+
Response to stress	GO:0006950	*RAD51AP1*	ENSMMUG00000015189	−	−

Cells filled with color (blue) represent gene expression that was negatively associated with a methylation difference.

## Data Availability

All WGBS data that support the findings of this study were deposited in the Sequence Read Archive (SRA) database under the accession code PRJNA866278. All reads of RNA sequences were unpublished data of the key laboratory of bio-resources and eco-environment of the Ministry of Education, College of Life Sciences, Sichuan University, China.

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
