# Peer review of "Genome-Wide DNA Methylation Profile Indicates Potential Epigenetic Regulation of Aging in the Rhesus Macaque Thymus"

_ijms, 2022, doi:10.3390/ijms232314984_

Round 1

Reviewer 1 Report (New Reviewer)

Authors presented WGBS and mRNA-seq analysis for two young and two adult rhesus macaques. They manuscript may help assist with understanding the mechanisms of aging thymus.

The introduction is sufficient.

The material and method is well designed. Authors used the right molecular and statistical analysis.

Authors clearly presented the results. The used figures and tables are sufficient.

I suggest that this manuscript should be accepted in the present form.

Author Response

Reviewer 2 Report (New Reviewer)

Round 2

Reviewer 2 Report (New Reviewer)

Minor comments

L114: Please correct in the legend of Fig. 1 the age for the adult individuals. Instead of “9 year old” it should be “9 years old”. It needs to be corrected in Fig. 1A and 1B.

L121-122: this sentence now does not make sense anymore. I suggest: “A total of 29,643 DMRs were identified from the thymus between the two age groups.”   The fact that they were differentially methylated is already included in the term “DMR”.

L189: not sure what is meant with “The average transcript per million (TPM) level of genes…”. The term “Transcripts Per Million (TPM)” describes a normalization method for RNA-seq, so it should be read like "for every million RNA molecules in the RNA-seq sample, x [here it is 6] came from this gene/transcript." To determine TPM, the raw count of reads mapped is divided by the transcript's length, giving a normalized transcript-level expression.

So, I assume, what the authors meant is “The average number of transcripts per million RNA molecules (TPM) was six:”

L211: please correct to “log2 mean expression”

L212: please correct to “log2-fold change” (instead of “log2Foldchange”)

L349: by mistake, the former correct “differentiate” was wrongly changed to “differentiation”.  Please reverse to “differentiate”.

Author Response

This manuscript is a resubmission of an earlier submission. The following is a list of the peer review reports and author responses from that submission.

Round 1

Reviewer 1 Report

In this manuscript, Qiu and colleagues reported an atlas of DNA methylome and transcriptome of thymus in young and old rhesus macaque. They found that most DMRs were hypermethylated in aged group and mainly located on introns. DEGs were enriched in immune system related processes. Next, they analyzed the correlation between the alterations of DNA methylome and transcriptome. They focused on the promoter methylation and found a slight decrease of methylation. Finally, they identified age-associated methylated sites and DMRs. However, I have some concerns on this study:

Major points: There were only 2 individuals for each group while the two individuals are different sex (one male, one female). It is hard to reach the conclusion that such differences is caused by aging but not by individual variation (for instance, many underlying diseases in old individuals) and sex bias.

Minor points: Some labels of figures weren’t clear enough for understanding.

Reviewer 2 Report

The manuscript titled “Genome-wide DNA Methylation Profile Indicates Potential Epigenetic Regulation of Aging in the Rhesus Macaque Thymus” by Qiu et al. did a multi-omics study on 4 thymus samples of 2 young and 2 aged non-human primate model species Rhesus Macaque. It is an interesting descriptive study which can be a good complement to the recent work published last year by Horvath et al. which analyzed the DNA methylation profile of 281 samples from the same species (10.1007/s11357-021-00429-8) and another preprint on 4 primates including Rhesus Macaque (10.1101/2020.11.29.402891), though I am a bit surprised that the authors did not mention them at all, not to say discuss their data in the manuscript. Additionally, I understand that such samples are not easily available for omics study, but it is still important to note that this study is limited by the lack of applicability of many statistical methods (with n=1 per age per sex, it is unrealistic to do many statistical analyses). Nevertheless, such descriptive study in a new organ related to immune functions is beneficial to the field, and I find the topic interesting and it is attractive to both experts and the general audience. Please see below a few suggestions which will hopefully help improve this manuscript.

1.      As there are only 4 samples, a simple PCA and correlation matrix for all the samples can be applied from all CpGs and/or CpGs from specific genomic regions. It will be important to compare individual differences in case it is larger than age-related differences.

2.      It has been a widely discussed topic that females' age slower epigenetically than males in mammalian species. I wonder if the authors could further analyze sex-specific age-related changes in the methylome. A Venn diagram separating Figure 4A by sex can be a good idea if possible.

3.      As the discussion of citation 34 (Horvath et al.) is based on the human methylation clock, it may be better to include the Horvath et al. Rhesus Macaque methylation study (doi: 10.1007/s11357-021-00429-8) and Horvath et al. pan-primate methylation clock (doi:10.1101/2020.11.29.402891) into the discussion in addition to Horvath et al., and make further analyses to their published data to allow direct comparison within the same species. It is important because these are not just the constructions of the aging biomarkers, but also detailed analyses of Rhesus Macaque DNA methylome from the aging perspective. Particularly, I wonder if their finding on Klf4 promoter is consistent with the results by the authors, and if the conclusions made by the authors can also be found in their data.

4.      Additional note regarding the previous point: Although WGBS provides high-coverage DNA methylome data to allow whole-genome analyses, most of the methylation studies that reveal age-related changes are done on either Illumina EPIC/450K microarray or HorvathMammalMethylChip40 microarray platform, with minor exceptions. Therefore, the authors can extend their methylation analyses to one of these platforms if possible, especially HorvathMammalMethylChip40 microarray platform where the articles I mentioned are based on. I assume that this will allow an analysis of the samples from the authors on the epigenetic aging biomarkers, and determine if these animals age faster or slower than usual. As it has been existing studies on other Rhesus Macaque tissue samples, it is also important to identify the thymus-specific age-related changes and make comparisons to see if the conclusions from both articles are consistent with each other.

5.      Despite the suggestion to the authors on analyzing the microarray data, it should be noted that microarrays have both advantages and disadvantages: One advantage WGBS has over microarrays is that it covers a much larger part of the genome. I noticed that the Horvath et al. pan-primate article mentioned the CpG methylation of polycomb repressive complex 2-targeting sites, which become increasingly methylated with age in humans and are hypothesized as a causal factor of age-related epigenetic changes. I wonder if an analysis on such sites can be analyzed in the data generated by the authors, as WGBS seems to have a natural advantage in such studies over Illumina microarrays which provide limited coverage of the genome.

Minor points:

1.        The x-axis label of Figure 2 A should be checked.

2.        The resolution of Figure 3A can be improved.

Round 2

Reviewer 1 Report

Thank the authors for their response and technical improvement. However, I don't think the authors have really addressed my concerns on their experimental design (only two samples for each group). 

Reviewer 2 Report

I see that the manuscript is much improved.

I do notice your point in the response to the other reviewer that individual differences and age differences are not the major differences in your paper. However, as you focus on the epigenetic regulation of ageing as mentioned in the title, I still feel that the current analysis in the context of ageing is shallow and that more evidence is needed, especially considering that multiple clocks are already available.

Particularly, I would like to clarify points 3,4 and 5: If the current platform limits the utilization of the clock to analyse age-related epigenetic regulations, it may be a good idea to apply the same DNA samples (or DNA samples from the same tissues) to the Horvath mammalian chip HorvathMammalMethylChip40. In this way, you may then be able to apply the Rhesus Macaque clock published by them and benefit from their large sample size to do a meta-analysis to counter the problem with small sample sizes. You are analysing a completely different tissue, so I do not think it will be a severe overlap between the topics. Besides, you can do a cross-platform cross comparison which will give more credibility to your findings. A secondary option is to utilize Illumina human chips such as EPIC or 450K. These are all commercially available microarrays widely used in mammalian samples, so I would not think it is totally unfeasible. There are only 4 samples to analyze after all.

Minor point: PCA and spearman correlation matrix can be listed in the main figure. Besides, font sizes are too small on the label inside the PCA plots.